# The Allocation Change of Rural Land Consolidation Type Structure under the Influence Factors of Different Geographical and Economic Development of China

**DOI:** 10.3390/ijerph20065194

**Published:** 2023-03-15

**Authors:** Qinglei Zhao, Guanghui Jiang, Mingzhu Wang

**Affiliations:** 1College of Geography and Tourism, Qufu Normal University, Rizhao 276826, China; zqlsdxt05@163.com; 2State Key Laboratory of Earth Surface Process and Resource Ecology, School of Natural Resources, Faculty of Geographical Science, Beijing Normal University, Beijing 100875, China

**Keywords:** land consolidation type structure, spatio-temporal change, driving mechanism, regional differences

## Abstract

Land consolidation structure reflects land consolidation function, and studies about its spatio-temporal change and driving mechanism can serve for regional management and control land consolidation. At present, the analysis of regional differences, time changes, and driving factors of land consolidation type structure change is relatively lacking. Based on the data of provincial acceptance projects from 2000 to 2014, this paper analyzes the spatio-temporal change of rural land consolidation type structure in China, discusses the impact of relevant policies, and identifies the socio-economic driving factors in key regions by employing correlation analysis and the PLSR (partial least squares regression) method. The results showed that from 2000 to 2014, the proportional increase of land arrangement in China was significantly correlated with the proportional decrease of land reclamation (R^2^ = 0.93), and the proportional decrease of land development (R^2^ = 0.99) showed an obvious co-evolution pattern of increase and decrease; *TI_LC_* (The area of land development/The area of land arrangement) decreased from 2.14 to 0.91 in 2002–2003. Since 2003, the dominant type of land consolidation in China has gradually changed from land development to land arrangement. However, the proportion of land development in QT (Qinghai-Tibet), JY (Jin-Yu), and FGH (Fujian-Guangdong-Hainan) areas is still more than 40%; the change of land consolidation type structure was influenced by policies, social and economic factors, such as urbanization rate, fixed assets investment, industrial proportion, and population density, and the regional difference was significant: the eastern section (JZS, Jiangsu-Zhejiang-Shanghai) is the industry proportion, the central area (HHAJ, Hunan-Hubei-Anhui-Jiangxi) is grain production and fixed assets investment, the western region (NW, Northwest China) has the urbanization rate, grain production, population density, and fixed assets investment. Land consolidation structure should be configured differentially in each region based on the identification of regional function orientation and comprehensive consideration of regional resource endowment and development needs and directions to improve the efficiency of land consolidation.

## 1. Introduction

Land consolidation in China has evolved from land arrangement, the definition of which is to adjust land relations and organize land use [1]. It is an effective means to alleviate the contradiction between people and land, maintain the balance of quantity and quality of cultivated land, and improve the supply of construction land. It is of great importance to ensure national food security, coordinate regional economic and social development, and realize the optimal allocation of land resources [1,2]. Since the Ministry of Land and Resources issued the first batch of state-invested land consolidation projects in 2001, land consolidation has gradually changed from spontaneous, disorderly, and unstable input to organized, standardized, and relatively stable input [2,3,4]. From 2000 to 2016, the total scale reached 1929.18 × 10^4^ ha, with an average annual growth rate of 50%. The implementation of a large number of land consolidation projects has increased land-use efficiency, improving regional production and living conditions as well as the ecological environment [5,6].

Land consolidation structure refers to the proportion of land arrangement, land reclamation, and land development projects [7,8]. Since 1999, various types of land consolidation in China have different functions: Land arrangement is an activity that adjusts the land’s form, land ownership, and land-use structure, improves and builds infrastructure facilities to increase land-use efficiency and productivity, and improves production, living, ecological conditions and functions, including agricultural land consolidation and construction land consolidation [9,10,11]. Land development is the process of making unused land available through engineering, biological, or comprehensive measures, including agricultural land development and construction land development [12]. Land reclamation is an activity that makes land destroyed due to production and construction activities and natural disasters available by taking comprehensive control measures (land consolidation terminology issued by the Ministry of land and resources in March 2018 (TD/T 1054-2018)).

In recent years, the central government and former Ministry of Land and Resources have issued a series of policies aimed at optimizing the land consolidation structure, adjusting its objectives and contents, enriching its means, and enhancing its functions [13,14]. The Land Management Law stipulates strict restrictions on the development of unused land. In the Interim Measures for the Projects Management of Land Development and Consolidation for State Investment promulgated in 2001 (No. 316 of Land Resources Development and Consolidation [2001]), the principles for determining the projects with land arrangement and land reclamation as the main tasks and the appropriate development of unused land are clearly defined. “Some Opinions on Land Development and Consolidation” (No. 363, 2003) called for land arrangement and reclamation as the focus. The “Decision of the Central Committee of the Communist Party of China on Several Major Issues Concerning Promoting Rural Reform and Development”, made in 2008, further emphasized land arrangement as the focus, land reclamation as the auxiliary, and land development as the supplement. In May 2014, the Regulations on the Economical and Intensive Use of Land issued and implemented by the Ministry of Land and Resources defined the denotation and objectives of land consolidation. The scope of land consolidation in China is no longer limited to agricultural land or rural land but has become the consolidation of the whole land [6,15].

The central government has given great policy support to the adjustment of land consolidation structure from the perspective of top-level design. This structural arrangement reflects the change of the contents and objectives of land consolidation, from developing new land to excavating stock land, from the pursuit of increasing cultivated land area, improving the quality of cultivated land, and adjusting the proportion of urban and rural construction land to optimizing land property relations, organizing land use, protecting the ecological environment, and promoting major national and regional strategy [16,17].

Land consolidation has gone through three stages of development, undergoing a new adjustment, to reach the ecological concept-oriented fourth stage [16]. Yan et al. have formed the overall design concept of the regional integrated land consolidation strategy model and the future land consolidation strategy [17]. Its essence is defined as “readjustment of man-land relationship”, and its function is analyzed as “satisfying people’s demand for space improvement of production, living and ecological”. Rural land consolidation should be upgraded to a national strategy and put forward a plan of action and implementation priorities to scientifically promote the rural land consolidation strategy [18]. Rural land consolidation is a systematic project which should be implemented by respecting local stakeholders’ willingness and requests [19]. The land consolidation system can be divided into three layers: the core layer, the supporting layer and the criteria of land consolidation, and the constructed framework of the land consolidation system under the logical thinking process of land use and cognition [20]. The interactions and collaborations between land administration and land consolidation is beneficial to face outlined new developments and current challenges [21,22,23].

The intensity, potential, and difficulty of land consolidation are the key indicators to evaluate coordination of agricultural land consolidation with the help of mechanical balance models and geospatial statistical analysis [24,25]. The logical framework of rural land consolidation can be constructed from five aspects: guiding ideology, target orientation, macrostrategy, overall planning, and arrangement of renovation projects and functions [26,27]. The concept of “anti-planning” for the global land consolidation process has also been proposed and studied [28]. It is proposed to use the synthetic LCR_i_ indicator to rank the type and urgency for implementation consolidation projects [29]. Agricultural productivity, disaster-bearing capacity, ecological environment, and economically viable and socially acceptable are all factors that need comprehensive consideration in land consolidation [23,30,31,32,33].

Land consolidation effects are mainly reflected in rural reconstruction, including the spatial structure of rural agriculture, industry, and service industry [34,35]. Scholars employ whole life circles (the social network model), niche theory, a hybrid method (a genetic algorithm and fuzzy logic techniques) to identify the implementation risks and impacts of major land consolidation projects [36,37,38]. Some scholars believe that priority should be given to land exchange followed by land consolidation [39], and they investigate the feasibility of land consolidation in the customary tenure by juxtaposing the local conditions of the study areas with the baseline conditions for land consolidation outlined in the literature [40,41,42,43]. They confirmed the importance of land consolidation processes not only for the organization and recovery of ownership and cadastral records but also for the improvement of agricultural use of landscape and protection of natural resources [21,44,45,46].

Different stages of socio-economic development need to match different land consolidation contents and types [47]. Different regions in China have different natural and economic and social development stages, different land-use directions, intensity, structure, and problems, resulting in different land consolidation demands and structural allocation [48,49], so regional differential land consolidation should be implemented. According to the land-use zoning in the general land-use planning outline, the existing land consolidation planning zones are divided into nine regions for project layout. The scale and structure of land consolidation in different regions should be arranged according to particular factors, which is an important issue concerning the effectiveness of land consolidation. It is of great significance to identify and analyze the influencing factors of each stage and region accurately and then to find out the driving mechanism by integrating multifactors and multidimensions and to implement the guidance of differentiation in accordance with the scientific land consolidation management system [50].

In summary, the current research on land consolidation mainly focuses on strategy, model, benefit evaluation, influencing factors, and the effects of land consolidation [42,43,44], but research on land consolidation type structure is relatively limited. The research object generally is agricultural land improvement, the scale is mostly provincial and county, the methods are traditional correlation analysis, principal component analysis, and multiple logistic regression [51]. Therefore, this paper takes regions as research scales, and land consolidation type structures are taken as research objects. Partial least squares regression (PLSR), which combines the characteristics of principal component analysis and multiple regression that can overcome the problem of multiple collinearities among variables [52,53,54], is seldom employed. In view of this, this study uses the data of land consolidation projects from 2000 to 2014 in various provinces, employs JMP, SPSS, and ArcGIS 10.0 software platforms for correlation analysis, VIP value calculation and map making, and analyzes the changes of various types of land consolidation in order to reflect the spatial and temporal characteristics of land consolidation structure and regional differences, and discusses the policy impacts. Identification of the influencing factors of temporal and spatial changes, studying the driving mechanism, and then revealing the characteristics of regional differences is the aim of this study.

## 2. Materials and Methods

### 2.1. Data Sources

According to the relevant research [7,11,46] combined with the characteristics of influencing factors and variable availability of land consolidation, 12 factors were selected from economic, social, and natural aspects for analysis (Table 1). The GDP per capita, GDP per square kilometer, and fixed assets investment reflect living standards of the people in the region, the level of economic benefits per unit area of land, and financial support capacity. Elevation, arable land area, water resources per capita, agricultural machinery power, total grain production, and primary industry rate reflect the natural conditions and terrain elements of a region, regional arable land resources endowment, and agricultural development foundation. Urbanization rate and population density reflect the degree of population aggregation to the city and the intensity of land use. The difference of new construction land-use fees reflects the difference of regional land consolidation policies. The above factors affect the land consolidation activities from different angles, which need further analysis. Land consolidation project data is from the “China Land and Resources Statistics Yearbook”. Natural, economic, and social data is from the “China Regional Economic Statistics Yearbook” and the “China Urban and Rural Construction Statistics Yearbook”. China’s vector map is from the National Bureau of Surveying and Mapping Geographic Information.

The content of land consolidation should be carried out in specific space. The national land consolidation plan is a macroscopic land consolidation policy, which divides China into nine land consolidation areas [7,52]. Land consolidation zoning is a comprehensive division of the region on the basis of considering the natural, social, and economic, land-use status, land-use issues, land-use direction, regional development strategy, and the integrity of provincial jurisdictional boundaries. The nine land consolidation zones in China are Northeast China (NE), Jin-Yu (JY), Beijing-Tianjin-Hebei-Shandong (BTHL), Jiangsu-Zhejiang-Shanghai (JZS), Fujian-Guangdong-Hainan (FGH), Northwest China (NW), Southwest China (SW), Hunan-Hubei-Anhui-Jiangxi (HHAJ), and Qinghai-Tibet (QT) (Figure 1). Northwest China includes Shaanxi, Gansu, Ningxia, Xinjiang, and Inner Mongolia. Southwest China includes Chongqing, Sichuan, Guizhou, Guangxi, and Yunnan. Northeast China includes Liaoning, Jilin, and Heilongjiang. These nine districts have different geographical backgrounds, economic development levels, and land consolidation policies. This paper takes the provincial region as the basic unit, studies the impact of land consolidation policy in different regions, and analyzes and discusses the changes of land consolidation structure and its influencing factors in 2000–2014 in China and in various regions.

### 2.2. Correlation Analysis

Pearson product-moment correlation coefficient (PCCs) is a statistical method which can quantitatively measure the correlation between variables. When calculating a sample, the Pearson correlation coefficient is determined by R value, which reflects the degree of linear correlation between two variables, and the range of R is [−1,1]. If r > 0, that indicates that two variables are positively correlated; when r = 1, that indicates that the variables are completely positive linear correlation; if R < 0, that indicates that the two variables are negatively correlated; when r = −1, that indicates that the variables are completely negative linear correlation. The greater the absolute value of r, the stronger the correlation; when r = 0, that indicates that there is no linear correlation between variables [10,20,25]. The R formula is as follows:(1)$$R=\frac{\sum \;\mathrm{XY}-\frac{\sum \;X\sum \;Y}{N}}{\sqrt{\left(\sum \;X^{2}-\frac{{\left(\sum \;X\right)}^{2}}{N}\right)\left(\sum \;Y^{2}-\frac{{\left(\sum \;Y\right)}^{2}}{N}\right)}}$$
where X is the set of X coordinates of all points, Y is the set of Y coordinates of all points, and N represents the total number of points.

Multivariate statistical analysis between the land consolidation scale change and the socio-economic variables was completed in SPSS26 software. During 2000–2015, the spatial distribution of land consolidation scale and the correlation between time-varying land consolidation scale and natural socio-economic driving factors in each region were characterized by Pearson correlation coefficient.

### 2.3. Transition Index

The so-called transformation refers to the fundamental transformation process of the structure, operation model, and people’s concept of things. Land-use transition was first proposed by Grainger of the University of Leeds in the United Kingdom in his study of land use in forestry-dominated countries. He assumed that most forestry countries would go through several stages of development: sustained deforestation until a new balance was reached between forestry and agriculture. Long et al. proposed the process of land-use transformation which includes the long-term and tendentious changes of land-use patterns. The fundamental change of land-use patterns or the change of direction and trend indicates the occurrence of land-use transformation in a certain period of time [55].

With the change of time and the development of society and economy, the connotation of land consolidation has been deepened, from quantity management to quality management to ecological management and protection, and from the extension of simply replenishing the quantity of cultivated land to the comprehensive land consolidation aimed at underutilized, improperly utilized, and overutilized [16,17,50]. Land development and land arrangement represent different connotative characteristics, so this paper refers to relevant viewpoints [35,45,50] and uses the ratio of land development and land arrangement project scale to measure the land consolidation type structure in China, so as to reflect the land consolidation function. In different time and space, the changes and differences of the land consolidation type structure reflect the objectives and contents of land consolidation. The evolution further indicates the change of land use form and the stage of economic and social development. Due to the relatively small proportion of land reclamation and changes, they are not considered (Figure 2). The calculation method and the theoretical model are as follows:(2)$$TI_{LC}=\frac{A_{LD}}{A_{LA}}\;\times 100\%$$
where *TI_LC_* = is the type index of land consolidation structure change; *A_LA_* is the land arrangement area; *A_LD_* is the land development area.

### 2.4. PLSR

In this paper, partial least squares regression is used to analyze the relationship between response variables (land consolidation scale) and socio-economic driving factors between 2000 and 2015. The interpretation degree of independent variables to dependent variables is demonstrated by the Variable importance of projection (VIP) in the model [54].

The VIP formula is as follows:(3)$$VIP_{j}={\left\{P\sum_{h=1}^{m}\sum_{k}R^{2}\left(y_{k},t_{h}\right)w_{hj}^{2}/\sum_{h=1}^{m}\sum_{k}R^{2}\left(y_{k},t_{h}\right)\right\}}^{1/2}$$
where *p* is the number of variables X (including GDP per capita, economic density, investment in fixed assets, total grain output, proportion of primary industry, population density, and urbanization rate), m is the total number of potential variables extracted from *x*, *K* is the number of response variables Y, $w_{hj}^{2}$ is the weight of $X_{j}$ in each $t_{h}$ component weight, $\sum_{k}R^{2}\left(y_{k},t_{h}\right)t_{h}$ is the square of the explanatory variance of $t_{h}$ to dependent variable Y, and VIPj is the measure of the total contribution of *X_j_* to the PLSR model. If the value of VIP is greater than 0.5, it indicates that the independent variable is more important to the dependent variable, and the larger the value of VIP is, the more important it is. VIP calculation in PLSR is implemented in JMP13 software.

## 3. Results

### 3.1. Temporal and Spatial Changes of Land Consolidation Type Structure

#### 3.1.1. Temporal Variation of Land Consolidation Structure

From 2000 to 2014, the proportion of land arrangement increased, while the proportion of land development and land reclamation decreased. Since 2001, when the Ministry of Land and Resources issued the first batch of state-invested land consolidation projects, the overall scale of land consolidation in China has shown an upward trend, from 40.04 × 10^4^ ha in the initial stage to 333.73 × 10^4^ ha in 2016 (land consolidation project data from the "China Land and Resources Statistics Yearbook"), with a total scale of 1929.18 × 10^4^ ha, an average annual increase of nearly 50% (Figure 3 and Figure 4). The total scale of land development was 291.26 × 10^4^ ha, accounting for 20.31%, which decreased from 62% at the beginning of the period to 11% in 2014, and declined nearly 3.7% annually; the total scale of land arrangement was 1065.93 × 10^4^ ha, accounting for 74.32%, from 23% at the beginning of the period to 88% at the end of the period, rising nearly 4.7% annually. The total scale of land reclamation was 77.03 × 10^4^ ha, accounting for 5.37% of the total land reclamation, which decreased from 16% at the beginning of the period to 2% and decreased by 1% annually.

There are three points worth noting. First, the change of the land arrangement proportion negatively correlated with the change of the land development proportion and land reclamation proportion, and the coefficients were −0.966 and −0.996 (*p* = 0.01), respectively, showing an obvious co-evolution pattern of increase and decrease. From 2002 to 2007, the structure of land consolidation showed drastic changes, and the proportion of land arrangement was rapidly rising, while the proportion of land development and land reclamation decreased sharply, with an average annual variation of 10%, −8%, and −2% respectively. Second, since 2005, the scale of land consolidation changed from slow increase to more violent fluctuations. Third, the rate of newly added cultivated land caused by land consolidation showed a downward trend, from 95.27% at the beginning of the period to 5.27% at the end of the period, with an average annual decrease of 6%. Thus, it can be seen that the structure and scale of land consolidation in China was constantly adjusted, and the dominant type changed from land development to land arrangement; at the same time, the goal of land consolidation changed from simplicity to comprehensiveness, from simply supplementing the amount of cultivated land to comprehensively improving the various types of inadequate utilization, improper utilization, and excessive utilization of the land.

The land consolidation structure index (*TI_LC_*) decreased from 2.74 in 2000 to 0.08 in 2015, with an average annual decrease of 0.18 (Figure 5). Among them, from 2.14 to 0.91 in 2002–2003, the decline was the biggest, and it was the turning point where the scale proportion of land arrangement projects exceeded land development. In 2005, the index was below 0.5, and then the downward trend gradually stabilized. This showed that the land consolidation structure in China has been changing from stable to drastic. Since 2003, the dominant role of land development was replaced by land arrangement, and the proportion of land arrangement is still increasing. With the decrease of reserve land resources, land development activities of unused land were gradually replaced by land arrangement of excavating stock. It was not only improvement of the utilization efficiency of land resources, but also reduction of the development intensity to a certain extent and protection of the ecological environment.

#### 3.1.2. Spatial Difference and Change of Regional Land Consolidation Structure

Figure 6 shows the spatial distribution and standard deviation of the land arrangement, land reclamation, and land development scale from 2000 to 2014. Overall, land arrangement was concentrated in the SW and NE, land reclamation was concentrated in the JZS area, while land development mainly was concentrated in the central and western regions: the NW, SW, and HHAJ areas. At the same time, the standard deviations of land arrangement scale and the newly cultivated land rate were gradually increasing, which indicated that the regional differences were more obvious, while the standard deviations of land reclamation and land development scale were gradually decreasing, which indicated that the regional differences were becoming smaller and smaller.

Figure 7 presents the scale structure of land consolidation in four periods during the years 2000–2014 in each region of China. It can be seen that the spatial distribution of the three types changed from being concentrated in one area to dispersed in three areas, gradually reflecting the difference of distribution and the spatial orientation of the various types of land consolidation: that is, the land development concentrated in the NW with large land area and relatively backward economic development, and the land arrangement and land reclamation relatively concentrated upon the JZS and BTHS with small land area and relatively developed economy.

Figure 8 shows the change of the proportion of regional land consolidation types. Overall, the proportion of land arrangement in China increased by nearly 5 percentage points annually from 2000 to 2014, while the proportion of land reclamation and land development decreased by about 1 and 4 percentage points annually, respectively.

In the three stages, the proportion of land arrangement at the national level showed an upward trend, and the proportion of land reclamation and land development showed a downward trend. Among them, the proportion of land arrangement increased by nearly 8 percentage points from 2000 to 2005, and land development decreased by nearly 7 percentage points, all of which were the largest in the past period.

From the regional point of view, the proportion of land arrangement in each region showed an upward trend, and the annual growth rates of the NE and SW regions were higher than the national level by 1 and 0.5 percentage points, respectively. On the contrary, the proportion of land reclamation in each region showed a downward trend, and the annual decreases of QT and JY were higher than 0.5 and 0.3 percentage points respectively. For the change of the proportion of land development, QT and JY showed an increasing trend, and other regions showed a downward trend. The annual average decreases of NE and SW were 2 and 1 percentage points higher than the national level, respectively. It can be seen that the regional proportion change of land arrangement and land reclamation was consistent with the national change direction, and the change of the QT and the JY area of land development proportion was contrary to the national change trend. On the one hand, the overall structure of regional land consolidation lacked local characteristics; on the other hand, it reflected that those individual regions did not fully meet the national requirements, reflected in certain regional differences.

Figure 9 shows the changing process of regional land consolidation structure in China. At the regional level, in 2000–2005 and 2005–2010, except for the QT, FGH, and SW regions, the proportion of land arrangement in other regions changed in the same direction as that in the whole country. In 2000–2005, the proportion of land arrangement increased and the proportion of land development declined the most in the NE region, which was about 5 and 6 percentage points higher than the national level, respectively. The biggest drop was 9 percentage points higher than the national level in QT. From 2005 to 2010, the proportion of land arrangement and land reclamation in the FGH area decreased the most, at 5 and 3 percentage points higher than the national level, respectively. The proportion of land development in QT area decreased the most, at 15 percentage points higher than the national level.

However, from 2010 to 2014, the direction of land consolidation structure change in each region was more diverse. The JY, BTHS, JZS, FGH, and HHAJ regional land arrangement proportion showed a downward trend; the JY, BTHS, JZS, FGH, HHAJ, and QT regional land development proportion showed an increasing trend; and the JY and JZS proportion of land reclamation increased. The difference in the direction and extent of change of the proportion of regional land consolidation structure reflected the heterogeneity of the content of land consolidation in different regions, and the structure of land consolidation was adjusted according to the different natural and economic conditions in different regions.

From Figure 10, we can see that the land consolidation structure indexes of most provinces were close to the whole country and the land arrangement was dominant. It shows that the overall land consolidation structure has been adjusted and optimized. At the same time, the indexes of some provinces were still higher than 1; Tibet was as high as 10, and Guangdong was as high as 4, far exceeding the national level. Tibet’s economic development is backward, and the land consolidation is in a low stage, mainly simple land development. Guangdong Province has a high level of economic development, and a large demand for construction land; in order to meet the demand, the development intensity of unused land in the region was relatively large. These areas are still dominated by land development, and there is much room for optimizing land consolidation structures.

### 3.2. Influence Factors of Land Consolidation Structure

The above analysis showed that significant adjustments have taken place in the scale and structure of land consolidation during the study period, the dominance from land development to land arrangement, and the rate of newly added cultivated land caused by land consolidation has declined year by year. The change of land consolidation structure in different regions was basically consistent with the change of the whole country from the initial stage, gradually showing diversification in direction and increased heterogeneity, which reflected the coupling process of land consolidation structure and regional development. The layout of the land consolidation structure gradually began to reflect more local characteristics and became more targeted and adaptable from the whole country.

During the study period, China was in a period of economic transition. Due to different development conditions and directions, policy factors were also analyzed in the introduction. Besides policies, the impacts of economic and social factors on the land consolidation structure were also prominent and different in each region. This paper used the correlation analysis method to analyze the effect degree of the main influencing factors of the layout of land consolidation structure in China, namely, R^2^ (Figure 11), and employed PLAR to analyze the relative influence of the main socio-economic factors through VIP value which caused the changes of land arrangement, land reclamation, and land development scale in China and regions from 2000 to 2014 (Figure 12).

#### 3.2.1. Overall Influencing Factors of Land Consolidation Structure

The spatial distribution of land arrangement was affected by total grain yield (R^2^ = 0.64) and cultivated land area (R^2^ = 0.71). The region has a good foundation for agricultural development, which led to the earlier and greater land use, but was also accompanied by unreasonable land use. The demand of regional land arrangement of agricultural and construction was higher than that of other regions. The spatial distribution of land reclamation was positively correlated with economic density (R^2^ = 0.55) and population density (R^2^ = 0.56). The higher the ground average GDP, the higher the population density. On the one hand, this showed that land-use efficiency was high, the regional economic level was high, and the population was dense; on the other hand, it showed that the potential of regional land resources was smaller, and the intensity of regional development occupancy and destruction was larger. Therefore, land reclamation can be carried out to remedy the land damaged in the process of production and construction activities, which creates a better means. The spatial distribution of land development was significantly related to fixed assets investment (R^2^ = 0.60) and grain output (R^2^ = 0.57). Regional cultivated land endowment was good, suitable cultivated land resources were also rich, agricultural land development potential and power was greater. The higher the regional fixed assets investment, the greater the indication of more capital construction and real estate projects. These have a larger demand for construction land and more land development activities.

#### 3.2.2. Influencing Factors of Land Consolidation Structure in Different Geographical and Economic Regions

There were obvious regional differences in the driving factors of land consolidation structure change. The results showed that the urbanization rate played a key role in driving the land arrangement scale changes in FGH, BTHS, and NE (VIP > 1.1), the land reclamation scale changes in BTHS, FGH, and JZS (VIP > 0.90), and the land development scale changes in QT, NE, and FGH (VIP > 1.2). China’s urbanization has led to the overall, not only regional, urban expansion. At the same time, when the expansion occupied a large number of forest land, wetland, and other land resources, the lack of reasonable planning and standard requirements resulted in inefficient and disorderly land use, which was the main driver of regional land consolidation of FGH, BTHS, NE, QT, JZS, and so on. Fixed assets investment had a significant impact on the scale of land arrangement, reclamation, and development in various regions, especially in the less developed regions as a whole. This showed that government investment was an important driving force for land consolidation.

However, in areas where the economy was relatively lagging, there was little demand for land use and insufficient desire for bottom-up land consolidation. It was also more necessary to provide full play of the government’s macropolicy role. The proportion of primary industry had an important impact on the scale of land reclamation and development in NW and QT and had an important impact on land arrangement in JZS and SW. The proportion of primary industry reflected the industrial structure of the region: the lower the proportion, the higher the industrial structure and the level of development. The impacted regions were distributed into two extremes: the most advanced and most backward regions of the economy. Population density played an important role in the change of land consolidation structure in the central and western regions and a relatively small role in the eastern region, which indicated that the role of population size in the less developed regions was greater than in the developed regions. In addition, per capita GDP and ground average GDP had a greater impact on land arrangement and land reclamation in the central and western regions and a greater impact on land development in the eastern region. Regional differences in economic and social factors were an important feature of land arrangement, reclamation, and development structure changes.

During the period of economic transformation, the leading socio-economic driving forces of land arrangement, land reclamation, and land development scale change in China were different in different regions. For the whole country and most regions, the diversified socio-economic factors comprehensively drove the changes of land consolidation scale and structure. Overall, the main socio-economic factors causing the scale change of land arrangement, reclamation, and development were urbanization rate, fixed assets investment, total grain output, the proportion of primary industry, and the population density (VIP > 0.8). At the same time, there were differences between the eastern, central, and western regions. This paper chose the JZS area to represent the eastern developed area, the HHAJ area to represent the central area, and the NW area to represent the western underdeveloped area. From the perspective of regional spatial differences and temporal changes, the correlation analysis of land consolidation scale and structure change and important influencing factors was carried out.

The results showed that in the JZS area, the change of land arrangement scale was only negatively correlated with the proportion of primary industry (R = −0.68); the change of the land reclamation scale had no significant correlation variables; and the land development scale was negatively correlated with population density (R = −0.52), fixed assets investment (R = −0.56), and total grain output (R = −0.57). In the HHAJ area, the change of the land arrangement scale was positively correlated with urbanization rate (R^2^ = 0.67), fixed asset investment (R^2^ = 0.69), and total grain output (R^2^ = 0.73) and negatively correlated with the proportion of primary industry (R = −0.61); there was no significant correlation between the changes of the land reclamation scale, and the land development scale was positively correlated with urbanization rate (R^2^ = 0.54) and grain production total output (R^2^ = 0.60). In the NW area, the change of the land arrangement scale was positively correlated with urbanization rate (R^2^ = 0.85), population density (R^2^ = 0.83), fixed asset investment (R^2^ = 0.91), and total grain output (R^2^ = 0.93) and negatively correlated with the proportion of primary industry (R = −0.69), while the change of the land reclamation scale was negatively correlated with urbanization rate (R = −0.53) and population density (R = −0.55), was positively correlated with the proportion of primary industry (R^2^ = 0.65), and there was no significant correlation between the change of the land development scale (Figure 12).

Comprehensive analysis showed that for different stages of economic development, the regional influence factors of land arrangement, reclamation, and development were different. In the economically and socially backward areas (NW), the intensity of construction land was relatively low, and the existing land supply sufficiently met the demand of urban land. Land consolidation mainly aimed at agricultural land. With the large-scale development of unused land, arable land reserve resources reduced. Farmland consolidation with the main purpose of improving the level of farmland infrastructure become the main body of regional land consolidation and has been deepened into high-standard farmland construction.

For the economic and social development of medium-sized areas (HHAJ), the regional economic and social development brought greater intensity of land use, the existing scale of land supply has difficulty meeting the needs of urban development, so the region adopted balancing between the increase and decrease of urban and rural construction land to improve the level of regional conservation and intensive land to support regional urban development and cultivated land scale.

In the economically and socially developed region (JZS), the level of economic and social development of the region made the demand for land more intensive, and both quantity and quality of land were pursued. Land development was no longer the main means; the region mainly implemented ecological farmland construction, repaired damaged land, improved the per capita environment, carried out comprehensive land consolidation within the scope of village consolidation, further enhanced the level of regional conservation and intensive land use, and promoted the development of green land consolidation.

## 4. Discussion

### 4.1. Change Mechanism of Land Consolidation Type Structure

Different stages of economic and social development correspond to a certain land-use pattern and a certain land-use transition stage. The position of the current land-use pattern of a certain land type in its entire transition stage can provide reference for the relevant departments in the formulation specific process of land consolidation objectives, models, and policies. Therefore, the differences of land-use patterns under the influence of different stages of social and economic development will lead to different patterns and contents of regional land consolidation [55,56]. Natural conditions and cultivated land resources endowment will affect the initial spatial distribution of land consolidation type structure, but its elasticity is small and hard to change. Relatively speaking, the change period of economic and social conditions with time was relatively short. Therefore, during the study period, the time change of land consolidation scale and structure was affected mainly by economic and social factors. According to the relevant research and the purpose of this paper, the social factors related to land consolidation can be reflected by policies, urbanization level, and so on. Economic factors can be measured by GDP per capita, fixed assets investment, industrial proportion, and so on. Based on the above analysis, this paper held the idea that land consolidation type structure was optimized by increasing the proportion of land arrangement. The structure is the form of the elements within the system; the function is how the system with this structure in a certain environment can play a role or ability. Structure constrains function, and function tests structure and reacts to structure. The two interact to improve the structure and function of the system [45] (Figure 13).

Similarly, the structure and function of land consolidation are mutually constrained. The identification of spatial distribution factors and the driving mechanism analysis of regional temporal change are of great significance to the implementation and function of land consolidation zoning [46]. The object of land consolidation should be changed from “single advance” to “comprehensive elements”, which should not be confined to a single cultivated land, collective construction land, or other single elements to achieve a comprehensive improvement of the seven elements of “mountains, water, fields, roads, forests, villages and cities”. The strategic task of the first stage of our land consolidation is cultivated land protection. Subsequently, farmland remediation with the main purpose of improving the level of farmland infrastructure began to become the main body of land consolidation in China. In 2004, a comprehensive land consolidation in the whole township and village area was carried out, which is the third stage of China’s land consolidation [16,17].

After the above three major adjustments, the objectives and tasks of China’s land consolidation have been diversified again, but compared with the initial stage, the connotation is clearer and the means are richer. At present, China’s land consolidation is in the fourth adjustment period, the main feature is green land consolidation [16].

In other words, the combined efforts of various departments such as land, agriculture commission, forestry, water conservancy, and environmental protection in the consolidation area are carried out to simultaneously promote the consolidation of mountains, water bodies, farmland, roads, forests, urban and rural residential areas, industrial and mining land, and other types. To achieve the goals of intensive production, quality of life, and ecological improvement, the scope of land consolidation should be changed from “project bearing” to “comprehensive coordination”. The transformation and development of land consolidation should change the rigid thinking that takes a specific individual project as the consolidation category, and turn to comprehensive planning, comprehensive design, and comprehensive consolidation. The regional differences and inter-relation of base areas focuses on giving full play to the comparative advantages of each region and promoting reasonable division of labor and cooperation among regions [17,50]. We should develop land use in all regions, prevent duplication of construction and convergence of industrial structure, and promote the coordination between regional economic, industrial, and population development and land use. At a more macrolevel, through establishing an international development perspective, land consolidation should be focused on regional and differentiated land consolidation. It is committed to safeguarding national strategies such as “One Belt, One Road”, “Beijing-Tianjin-Hebei”, and “Yangtze River Economic Belt” and promoting structural optimization and spatial coordination of land resources in metropolitan areas, city clusters, and integrated regions [3,17].

In December 2019, on the basis of summarizing the experience of local pilot projects for comprehensive land consolidation, the Ministry of Natural Resources issued Notice of the Ministry of Natural Resources on Carrying out Pilot Projects for Comprehensive Land Consolidation (Natural Resources Development (2019) No. 194), which proposed that towns and villages should be the basic implementation units on the premise of scientific and reasonable planning. We will improve the overall consolidation of agricultural and construction land and the protection and restoration of the rural ecology, improve the spatial patterns of production, living, and ecology, promote the protection of cultivated land and the intensive and economical use of land, improve the living environment in rural areas, and promote all-round rural revitalization [7,13].

### 4.2. Policy Implication

Generally speaking, the land-use pattern of China in the past 30 years to meet the needs of urban development by occupying a large number of agricultural land and ecological land was unsustainable. It regarded pursuit of economic interests as the main direction of land use. At present, China’s economic development is in a transitional period, and the focus of land use and land policy formulation is shifting from increasing land production and economic value solely to improving land ecological function and potential ecological value, from large-scale urban expansion to more intensive and sustainable development [3].

Since the beginning of the 21st century, China’s economy has been growing rapidly and its comprehensive national strength has been continuously enhanced. Based on the reality of China’s urban and rural development, the central government has begun to make major adjustments to the relationship between urban and rural areas. The basic conditions for the transformation of urban–rural relations are in place [3,8]. In this stage, on the basis of the continuation and deepening of the previous stage of reform, the focus of reform gradually expanded from breaking the dual economic system to the field of society, and the government’s direct input became the main means to adjust the urban–rural relationship. The most striking feature of this period is that the government improved the relationship between urban and rural areas by increasing direct investment in agriculture and rural areas. Since 2000, China’s urbanization has entered a fast lane. The urbanization rate exceeded 50% in 2011, and the per capita GDP exceeded USD 5000; the government has gradually carried out the reform of rural taxes and fees, trying to reduce the tax burden of farmers from the system and improve the relationship between urban and rural areas. In 2002, the report of the 16th CPC National Congress clearly took “coordinating urban and rural economic and social development” as the basic policy to solve the problem of urban–rural dual structure. The government has a clearer idea of breaking the urban–rural dual structure and promoting the “integration of urban–rural development” has become the new goal of building a new urban–rural relationship [8,27].

With the development of this socio-economy, land consolidation has been constantly adjusted. At present, it is in the transitional period from regional land consolidation to green land consolidation. The connotation has changed from solely increasing the amount of cultivated land to synthetically increasing the amount of cultivated land, improving the quality of cultivated land, and improving the ecological environment. In terms of means, it has changed from taking the project as the carrier to using the project and engineering as the carrier in combination with the Hook of Urban Construction Land Increase and rural residential land decrease, alongside the reclamation and utilization of abandoned industrial and mining land. In terms of content, it has changed from agricultural land consolidation to rural construction land, urban construction land, unused land development, and land reclamation [15,16,50].

Therefore, it can promote the transformation of land use and economic development relying on land consolidation. In the future, land consolidation should be based on the identification of regional functional orientation and comprehensive consideration of regional development needs and directions to differentiate the arrangement of the land consolidation structure and improve the comprehensive benefits of land consolidation.

Generally speaking, in large urban agglomerations and economically developed areas, urban multifunctional land consolidation methods should be explored to enhance the comprehensive service capacity of the land. In rural areas, rural production and living conditions and human settlements should be improved for the purpose of upgrading the level of rural infrastructure, guiding rural residents to live in concentrated areas, and establishing a pattern of rural land use [56,57]. In special areas connecting urban and rural space, the agricultural landscape construction and ecological construction should be strengthened in order to form a regional character bureau with high efficiency, beautiful scenery, and strong self-regulation ability. In mountainous and hilly areas, the efficient agricultural system of dry farming should be rationally developed and the land ecological integration should be carried out to improve the self-healing ability of degraded land ecosystems.

This study also has shortcomings: the data used in the analysis is not new enough, and does not include the last three years; the correlation analysis selected indicators are not comprehensive enough and may neglect other indicators that have an impact on land consolidation; the influencing factors zoning scale is to provinces and regions, relatively macro, and regional differences are still large. In view of the above problems, further research will be better explored.

## 5. Conclusions

This study uses the data of land consolidation projects from 2000 to 2014 in various provinces, employs multiple methods, analyzes the changes of various types of land consolidation, reflects the spatial and temporal characteristics of land consolidation type structure and regional differences, and discusses the policy impacts while studying the driving mechanisms.

Since 2003, the dominant land consolidation type structure in China gradually changed from land development to land consolidation. *TI_LC_* changed from 2.14 to 0.91 in 2002–2003, but the proportion of land development in the QT, JY, and FGH areas was still more than 40%.

From 2000 to 2014, the proportional increase of land arrangement in China was significantly correlated with the proportional decrease of land reclamation (R^2^ = 0.93) and the proportional decrease of land development (R^2^ = 0.99), showing an obvious co-evolution pattern of increase and decrease. The spatial distribution of scale was affected by cultivated land resources and grain production. The annual growth rate of SW and NW was the largest, at 40% and 20%, respectively. The spatial distribution of intensity was affected by the average GDP and population density. The intensity of JZS and BTHS was 6 and 4 times the national level, respectively.

The structural change of land consolidation was influenced by policies both social and economic. Quantitative analysis of socio-economic factors showed that the change of structure was driven by urbanization rate, fixed assets investment, industrial proportion, and population density and that the regional differences are significant: the eastern area (JZS) was the industrial proportion, the central area (HHAJ) was the grain output and fixed assets investment proportion, and the western area (NW) was the urbanization rate, grain output, population density, and fixed assets investment proportion.

## Figures and Tables

**Figure 1 ijerph-20-05194-f001:**
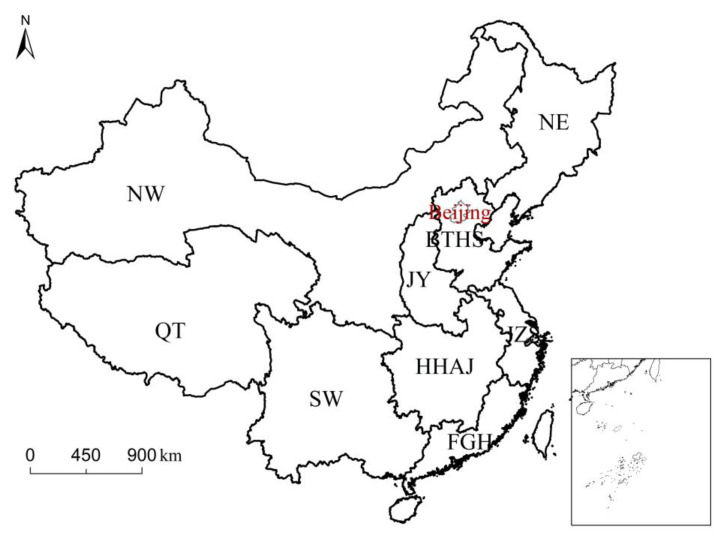
Study area. (NE: Northeast Region, JY: Jin Yu Region, BTH: Beijing Tianjin Hebei Region, FGH: Fujian Guangdong Hainan Region, JZS: Jiangsu Zhejiang Shanghai Region, NW: Northwest Region, SW: Southwest Region, HHAJ: Hunan Hubei Anhui and Jiangxi Region, and QT: Qinghai Tibet Region).

**Figure 2 ijerph-20-05194-f002:**
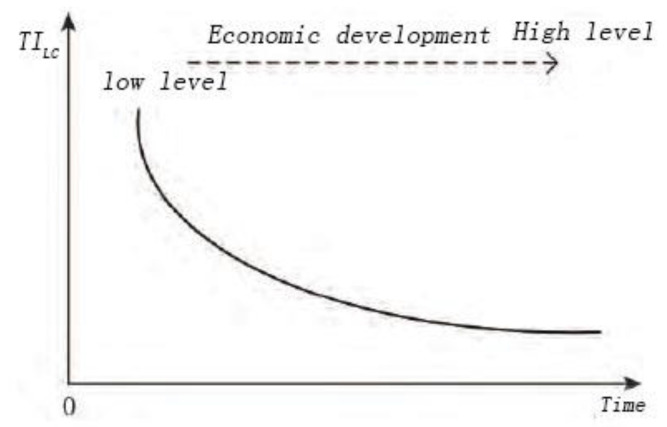
Theoretical model of land consolidation type structure change.

**Figure 3 ijerph-20-05194-f003:**
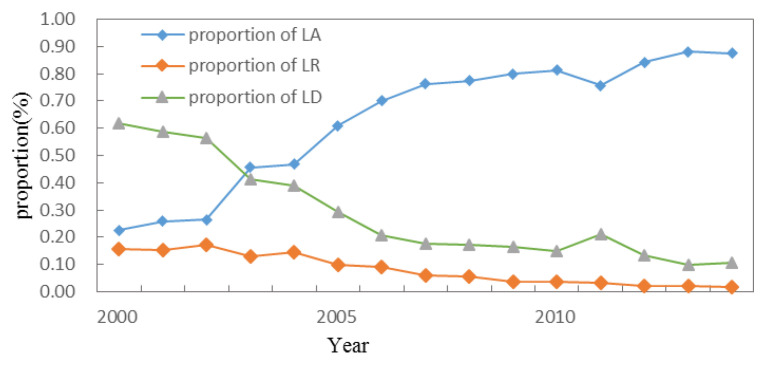
The structure change of LC in 2000–2014 in China (LA: land arrangement, LR: land reclamation, LD: land development).

**Figure 4 ijerph-20-05194-f004:**
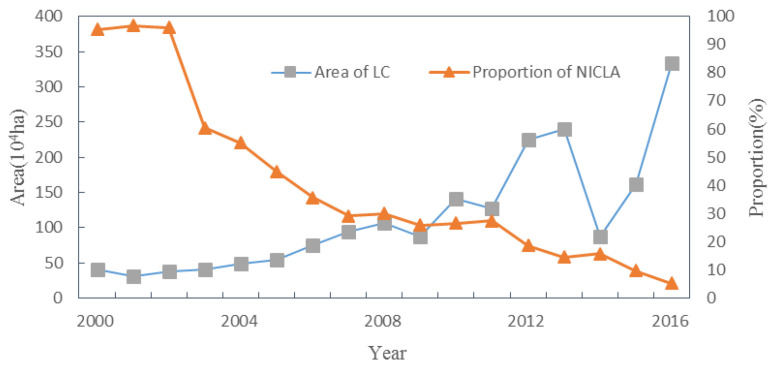
The change trend of LC area and NICLA proportion in China in 2000–2016. (LC: land consolidation, NICLA: newly increased cultivated land area; proportion of NICLA = NICLA/area of LC.).

**Figure 5 ijerph-20-05194-f005:**
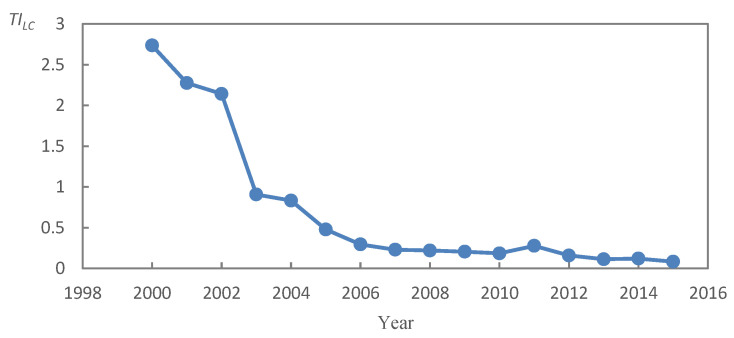
Interannual variation of *TI_LC_* in China ($TI_{LC}=\frac{A_{LD}}{A_{LA}}\;\;$× 100%, *TI_LC_*, structural change index for land consolidation; $A_{LA}$, land arrangement area; $A_{LD}$, land development area).

**Figure 6 ijerph-20-05194-f006:**
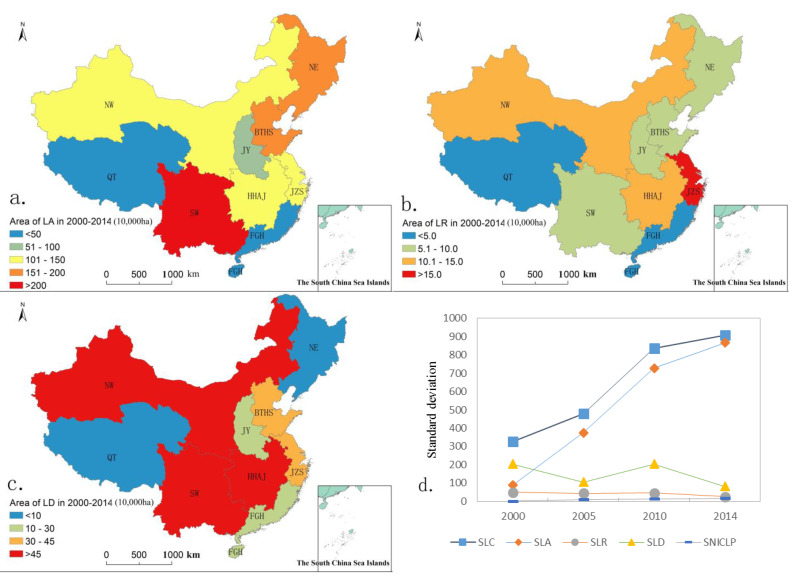
Geographic distribution of area of LA, LR, and LD in China in 2000–2014 and standard deviation in nine regions in 2000, 2005, 2010, and 2014 (Subfigures (**a**–**c**) is the geographic distribution of area of LA, LR, LD in China in 2000–2014, respectively; subfigure (**d**) is the geographic distribution of standard deviation of SLC, SLA, SLR, SLD, SNICLP in China in 2000–2014. SLC: standard deviation of land consolidation area, SLA: standard deviation of land arrangement area, SLR: standard deviation of land reclamation area, SLD: standard deviation of land development area, SNICLP: standard deviation of newly increased cultivated land proportion.).

**Figure 7 ijerph-20-05194-f007:**
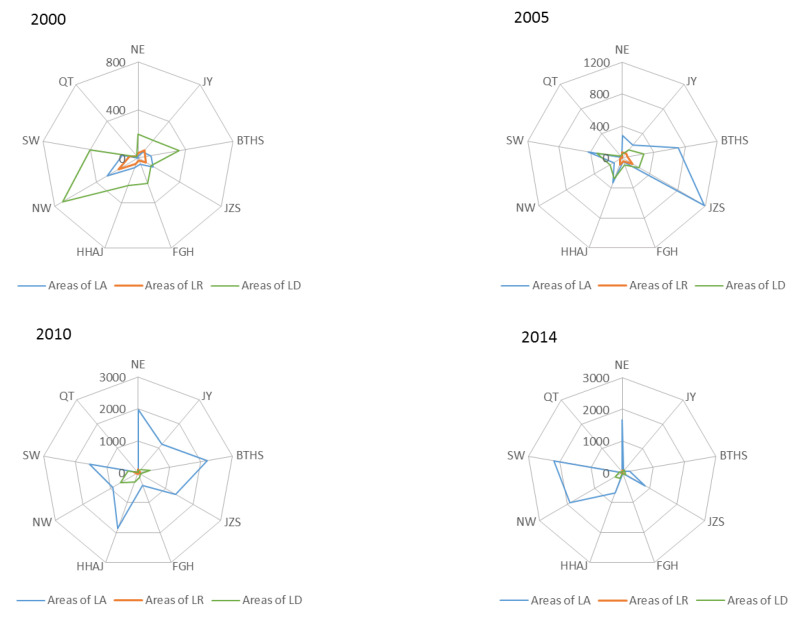
The structure change of LC in different regions of China from 2000 to 2014 (LC: land consolidation, LA: land arrangement, LR: land reclamation, LD: land development, NE: Northeast Region, JY: Jin Yu Region, BTH: Beijing Tianjin Hebei Region, FGH: Fujian Guangdong Hainan Region, JZS: Jiangsu Zhejiang Shanghai Region, NW: Northwest Region, SW: Southwest Region, HHAJ: Hunan Hubei Anhui and Jiangxi Region, and QT: Qinghai Tibet Region).

**Figure 8 ijerph-20-05194-f008:**
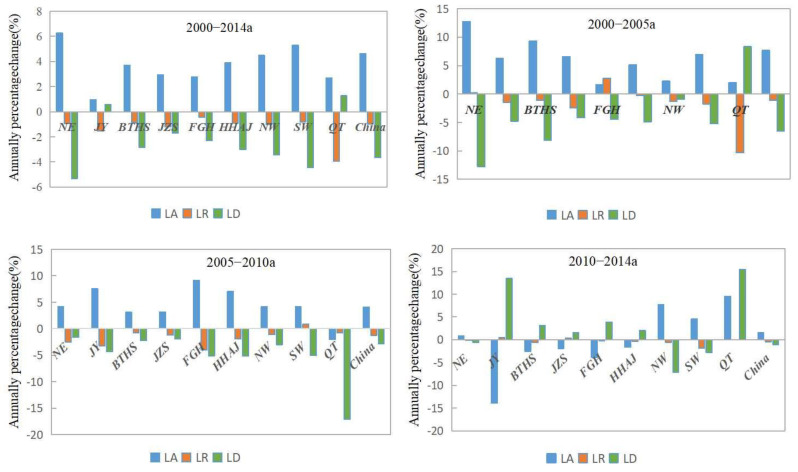
Annual percentage change in LA, LR, LD in each region of China in the periods of 2000–2014, 2000–2005, 2005–2010, and 2010–2014. (LA: land arrangement, LR: land reclamation, LD: land development, NE: Northeast Region, JY: Jin Yu Region, BTH: Beijing Tianjin Hebei Region, FGH: Fujian Guangdong Hainan Region, JZS: Jiangsu Zhejiang Shanghai Region, NW: Northwest Region, SW: Southwest Region, HHAJ: Hunan Hubei Anhui and Jiangxi Region, and QT: Qinghai Tibet Region).

**Figure 9 ijerph-20-05194-f009:**
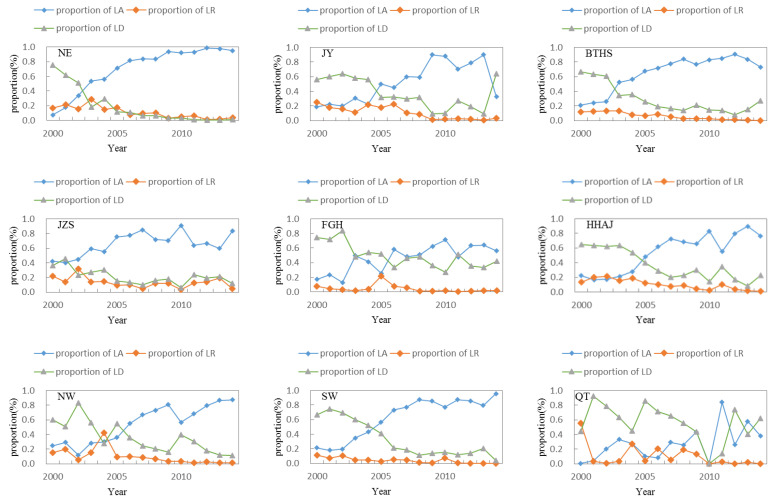
The structure change of LC in 2000–2014 in nine regions of China (LC: land consolidation, LA: land arrangement, LR: land reclamation, LD: land development, NE: Northeast Region, JY: Jin Yu Region, BTH: Beijing Tianjin Hebei Region, FGH: Fujian Guangdong Hainan Region, JZS: Jiangsu Zhejiang Shanghai Region, NW: Northwest Region, SW: Southwest Region, HHAJ: Hunan Hubei Anhui and Jiangxi Region, and QT: Qinghai Tibet Region).

**Figure 10 ijerph-20-05194-f010:**
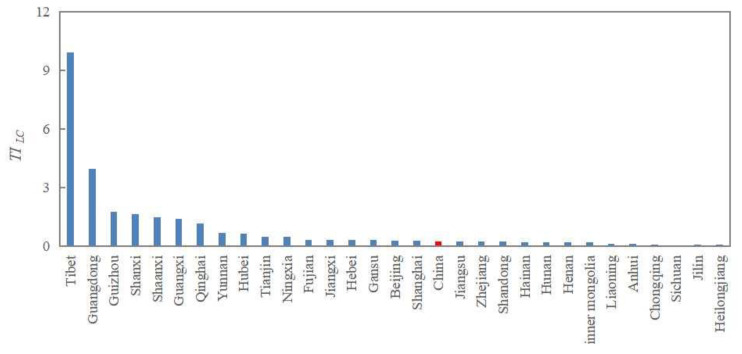
Provincial distribution of *TI_LC_* in China ($TI_{LC}=\frac{A_{LD}}{A_{LA}}\;$× 100%, *TI_LC_*, structural change index for land consolidation; $A_{LA}$, land arrangement area; $A_{LD}$, land development area).

**Figure 11 ijerph-20-05194-f011:**
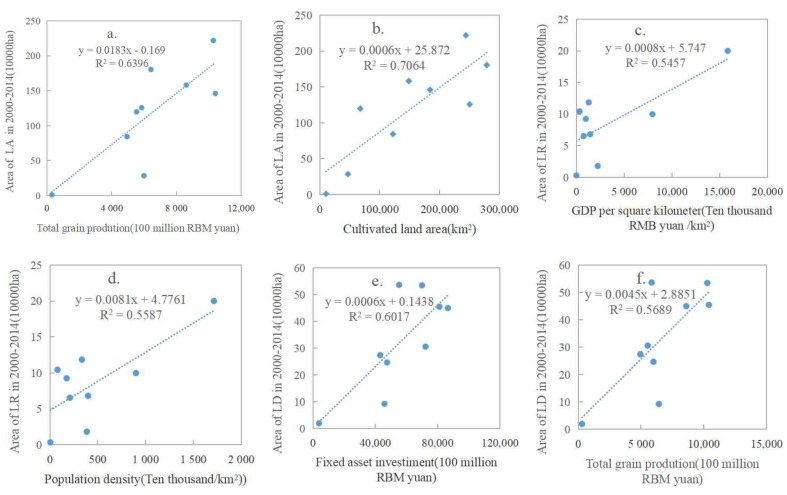
Scatter map of area of regional LA (**a**,**b**), LR (**c**,**d**), LD (**e**,**f**) and typical socio-economic and geographical indicators.

**Figure 12 ijerph-20-05194-f012:**
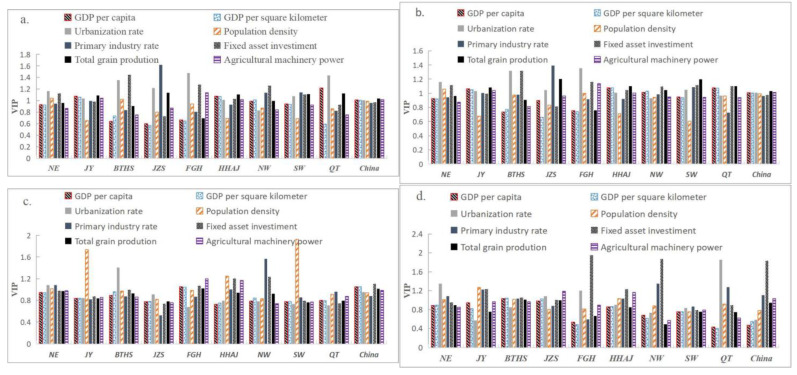
Driving factors of change in LC (**a**), LA (**b**), LR (**c**), and LD (**d**) areas indicated by VIP (Variable importance of projection).

**Figure 13 ijerph-20-05194-f013:**
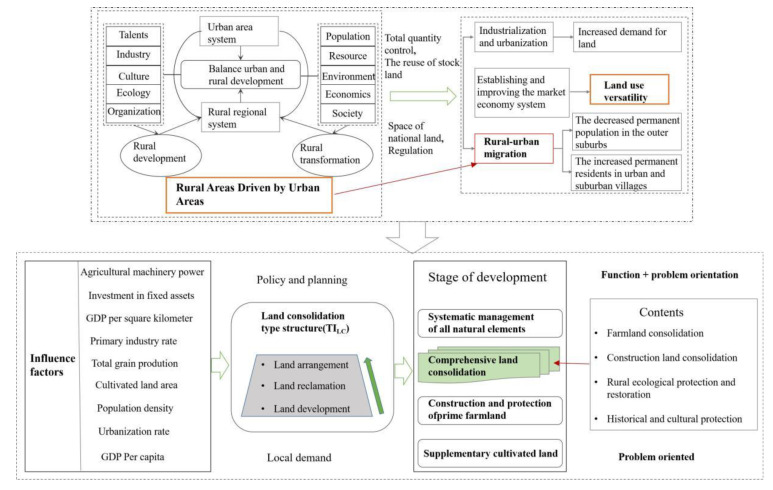
Causal-loop and evolution diagram of land consolidation.

**Table 1 ijerph-20-05194-t001:** Variables for calculating VIP value in PLSR analysis in nine land-use zoning regions (Figure 1).

Num.	Independent Variable	Calculation	Significance
1	GDP per capita	GDP/population	People’s standard of living
2	GDP per square kilometer	GDP/Land area	Reflects the efficiency of land use
3	Fixed asset investment	-	Financial support capacity
4	Total grain production	-	Arable land resource
5	Primary industry rate	Output value of primary industry/GDP	Basis of agricultural development
6	Urbanization rate	Urban population/total population	Population to urban agglomeration
7	Population density	Population/land area	
8	Elevation	-	-
9	Water resources per capita	-	-
10	Cultivated land area	-	-
11	Agricultural machinery power	-	-
12	New construction land use fees		

## Data Availability

Not applicable.

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
