# Peer review of "The Allocation Change of Rural Land Consolidation Type Structure under the Influence Factors of Different Geographical and Economic Development of China"

_ijerph, 2023, doi:10.3390/ijerph20065194_

Round 1

Reviewer 1 Report

The subject of the study is interesting and topical, with high scientific and practical importance.

The introduction is presented correctly, in accordance with the subject. Numerous scientific articles, in concordance to the topic of the study, were consulted.

Methodology of the study was clearly presented, and appropriate to the proposed objectives.

The obtained results are important and have been analyzed and interpreted correctly, in accordance with the current methodology.

The scientific literature, to which the reporting was made, is recent and representative in the field.

The following aspects are brought to the attention of the authors.

1.Grammatical errors (in the content of the article)

Eg

Page 3, row 98

"requests" instead of "request"

Page 3, row 119

"summed" instead of "sumed"

Page 3, row 127

"fragment" instead of "fragmentment"

Page 3, row 142, 143

"a genetic algorithm/ genetic algorithm" instead of " genetic algorithm "

Page 4, row 187

"collinearities" instead of " collinearity "

Page 6, row 258

"society" instead of " social "

Page 8, row 317

"showing" instead of " showed "

Page 8, row 323

"showed" instead of " shown "

Page 9, row 348

"was concentrated in" instead of " concentrated in "

Page 9, row 351

"were" instead of " was "

Page 11, row 408

"meet" instead of "met "

Page 11, row 409

" certain regional differences " instead of " a certain regional differences "

Page 12, row 422

"was" instead of " were "

Page 12, row 422,425,427

"the most" instead of " most "

Page 12, row 442

"was" instead of " were "

Page 12, row 447

"and a" instead of ", a "

Page 13, row 457

"showing" instead of "showed"

Page 13, row 487,488

"were" instead of "was"

Page 14, row 496

"were" instead of "was"

Page 14, row 503

"resulted" instead of " resulting"

Page 14, row 523

"were" instead of "was"

Page 16, row 523

" correspond" instead of " corresponding"

Page 16, row 591

"was affected" instead of " affected"

Page 16, row 599

"was embodied" instead of " embodied"

Page 16, row 603

"a" instead of " the"

Page 17, row 639

"areas" instead of "area"

Page 17, row 651

"showing" instead of "showed"

Page 18, row 667

"and may" instead of "may"

Page 18, row 672

"china" instead of "China"

It is recommended to check and correct the paper, as appropriate.

2.It is recommended to check and correct some sentences.

Eg

Page 12, row 429,430

“However, from 2010 to 2014, the inconsistent places where proportion of regional land consolidation structure change direction and the national got more,”

Page 13, row 488,489

“The higher the investment in regional fixed assets, indicating that the more capital construction, real estate projects, has a larger demand for construction land and more land development activities.”

3.Literature review should be line with research questions.

4.Policy implication should be line with results and discussion.

Author Response

Manuscript ID: ijerph-2169335

Title: How does the land consolidation structure of China change under economic development in transformation period?

Journal: International Journal of Environmental Research and Public Health;

Section “Environmental Health”; 

Special Issue “Urban Land Use Planning and Evaluation of Environmental Carrying Capacity”.

List of Changes

First of all, we appreciate your careful review and positive comments! All of your comments are very important and really helpful to revise and improve our manuscript (MS). According to your comments, we have made relevant changes (words in red) to the whole MS.The detailed revisions are listed below, responding to your comments point by point.

  1. 1. Replies to reviewer #1

 Comments and Suggestions for Authors

The subject of the study is interesting and topical, with high scientific and practical importance.

The introduction is presented correctly, in accordance with the subject. Numerous scientific articles, in concordance to the topic of the study, were consulted.

Methodology of the study was clearly presented, and appropriate to the proposed objectives.

The obtained results are important and have been analyzed and interpreted correctly, in accordance with the current methodology.

The scientific literature, to which the reporting was made, is recent and representative in the field.

The following aspects are brought to the attention of the authors.

Point 1: Grammatical errors (in the content of the article)

Eg

Page 3, row 98

"requests" instead of "request"

Page 3, row 119

"summed" instead of "sumed"

Page 3, row 127

"fragment" instead of "fragmentment"

Page 3, row 142, 143

"a genetic algorithm/ genetic algorithm" instead of " genetic algorithm "

Page 4, row 187

"collinearities" instead of " collinearity "

Page 6, row 258

"society" instead of " social "

Page 8, row 317

"showing" instead of " showed "

Page 8, row 323

"showed" instead of " shown "

Page 9, row 348

"was concentrated in" instead of " concentrated in "

Page 9, row 351

"were" instead of " was "

Page 11, row 408

"meet" instead of "met "

Page 11, row 409

" certain regional differences " instead of " a certain regional differences "

Page 12, row 422

"was" instead of " were "

Page 12, row 422,425,427

"the most" instead of " most "

Page 12, row 442

"was" instead of " were "

Page 12, row 447

"and a" instead of ", a "

Page 13, row 457

"showing" instead of "showed"

Page 13, row 487,488

"were" instead of "was"

Page 14, row 496

"were" instead of "was"

Page 14, row 503

"resulted" instead of " resulting"

Page 14, row 523

"were" instead of "was"

Page 16, row 523

" correspond" instead of " corresponding"

Page 16, row 591

"was affected" instead of " affected"

Page 16, row 599

"was embodied" instead of " embodied"

Page 16, row 603

"a" instead of " the"

Page 17, row 639

"areas" instead of "area"

Page 17, row 651

"showing" instead of "showed"

Page 18, row 667

"and may" instead of "may"

Page 18, row 672

"china" instead of "China"

It is recommended to check and correct the paper, as appropriate.

Response 1: Thank you very much for valuable comments. 

We have modified Grammatical errors (in the content of the article) as required.

We checked and modified similar errors throughout the article.

Point2: It is recommended to check and correct some sentences.

Eg

Page 12, row 429,430

“However, from 2010 to 2014, the inconsistent places where proportion of regional land consolidation structure change direction and the national got more,”

Page 13, row 488,489

“The higher the investment in regional fixed assets, indicating that the more capital construction, real estate projects, has a larger demand for construction land and more land development activities.”

Response 2:  We have checked and corrected some sentences, which not clearly expressed as above.

Point3: Literature review should be line with research questions.

Response 3: We have reorganized the total literature and put forward research questions based on it, which make the literature review line with research questions.

Point4: Policy implication should be line with results and discussion.

Response 4: Based on the findings of the paper, we have revised the discussion and policy implication, and cited the relevant literature, enriched the comparative analysis with research status.

The object of land consolidation should be changed from "single advance" to "comprehensive elements". The object of land consolidation should not be confined to a single cultivated land, collective construction land and other single elements. To achieve a comprehensive improvement of the seven elements of "mountains, water, fields, roads, forests, villages and cities". The strategic task of the first stage of our land consolidation is cultivated land protection. Subsequently, farmland remediation with the main purpose of improving the level of farmland infrastructure began to become the main body of land consolidation in China. In 2004, a comprehensive land consolidation in the whole township and village area was carried out, which is the third stage of China's land consolidation[16,17].

After the above three major adjustments, the objectives and tasks of China's land consolidation have been diversified again, but compared with the initial stage, the connotation is clearer and the means are richer. At present, China's land consolidation is in the fourth adjustment period, the main feature is green land consolidation[16].

Fig.13. Causal-loop and evolution diagram of land consolidation

Since the beginning of the 21st century, China's economy has been growing rapidly and its comprehensive national strength has been continuously enhanced. Based on the reality of China's urban and rural development, the central government has begun to make major adjustments to the relationship between urban and rural areas. The basic conditions for the transformation of urban-rural relations are in place[3,8]. In this stage, on the basis of the continuation and deepening of the previous stage of reform, the focus of reform gradually expanded from breaking the dual economic system to the field of social society, and the government's direct input became the main means to adjust the urban-rural relationship. The most striking feature of this period is that the government improved the relationship between urban and rural areas by increasing direct investment in agriculture and rural areas. Since 2000, China's urbanization has entered a fast lane. The urbanization rate exceeded 50% in 2011, and the per capita GDP exceeded US $5,000. the government has gradually carried out the reform of rural taxes and fees, trying to reduce the tax burden of farmers from the system and improve the relationship between urban and rural areas. In 2002, the report of the 16th CPC National Congress clearly took "coordinating urban and rural economic and social development" as the basic policy to solve the problem of urban-rural dual structure. The government has a clearer idea of breaking the urban-rural dual structure, and promoting the "integration of urban-rural development" has become the new goal of building a new urban-rural relationship[8,27].

Finally, special thanks to you for your constructive comments again!

Reviewer 2 Report

Manuscript “How does the land consolidation structure of China change under economic development in transformation period?”(ijerph-2169335)

The research design of this article is appropriate. This article selects the land consolidation structure and explores its spatial and temporal changes and driving factors, which has certain value. In terms of research methodology, the paper adopts correlation analysis and least squares estimation, which is innovative compared with previsous studies. My main concerns are provided as follow and these issues should be improved before paper acceptance:

1. Article structure, in the section “4.1 results and discussion”, 4.1 explores the spatial and temporal changes, so the titles of 4.1.1 and 4.1.2 should logically be the analysis of temporal changes and spatial changes, respectively; 4.2 explores influencing factors, where the title of 4.2.1 is economic and social factors, and the 4.2.2 explore the driving forces, including the urbanization rate, the government macro policies, etc..It  seems to be a repetition of 4.2.1. It is suggested to reorganize the content of section “4 Results and Discussion”.

2. The title of the article mentions the transition period of economic development, but the introduction section only introduces the background of land consolidation transition. It is suggested that the introduction section should add macroscopic background content so as to fit the title.

3. Literature review section, lacks a critical review of literature. Literature review should be concise rather than simply enumerating the literature.

4. 12 factors were selected from ecological, social and natural aspects, but didn't clarify why these factors are selected, are there any references to support this choice?

5. 3.1 data source section, the article lists nine land consolidation areas in China as the research area. A location map should be provided here with the location of the study area indicated.

6. Regarding the methodology, the 3.3 transition index section states that the article uses the ratio of land development and land consolidation projects to measure the structure of land consolidation in China and reflect the function of land consolidation. Is there any literature to support this? If yes, please cite it.

7. Figures: the figures resolution in part 4.1.2 is not enough resulting in blurriness and needs to be adjusted.

Author Response

Manuscript ID: ijerph-2169335

Title: How does the land consolidation structure of China change under economic development in transformation period?

Journal: International Journal of Environmental Research and Public Health;

Section “Environmental Health”; 

Special Issue “Urban Land Use Planning and Evaluation of Environmental Carrying Capacity”.

List of Changes

First of all, we appreciate your careful review and positive comments! All of your comments are very important and really helpful to revise and improve our manuscript (MS). According to your comments, we have made relevant changes (words in red) to the whole MS.The detailed revisions are listed below, responding to your comments point by point.

  1. 2. Replies to reviewer #2

 Comments and Suggestions for Authors

Manuscript “How does the land consolidation structure of China change under economic development in transformation period?”(ijerph-2169335)

The research design of this article is appropriate. This article selects the land consolidation structure and explores its spatial and temporal changes and driving factors, which has certain value. In terms of research methodology, the paper adopts correlation analysis and least squares estimation, which is innovative compared with previsous studies. My main concerns are provided as follow and these issues should be improved before paper acceptance:

Point 1: Article structure, in the section “4.1 results and discussion”, 4.1 explores the spatial and temporal changes, so the titles of 4.1.1 and 4.1.2 should logically be the analysis of temporal changes and spatial changes, respectively; 4.2 explores influencing factors, where the title of 4.2.1 is economic and social factors, and the 4.2.2 explore the driving forces, including the urbanization rate, the government macro policies, etc..It  seems to be a repetition of 4.2.1. It is suggested to reorganize the content of section “4 Results and Discussion”.

Response 1: Thank you very much for valuable comments. 

We have reorganized the content of section “4 Results and Discussion”. Because part two is merged into part one, so this partial order becomes “3 Results and Discussion”. The 3.1.1. is modified to “Temporal variation of land consolidation structure”; the 3.2.1. is modified to “Overall influencing factors of land consolidation structure”; the 3.2.2. Influencing factors of land consolidation structure in different geographical and economic regions. At the same time, we have adjusted and merged the two parts of the content.

Point 2: The title of the article mentions the transition period of economic development, but the introduction section only introduces the background of land consolidation transition. It is suggested that the introduction section should add macroscopic background content so as to fit the title.

Response 2: We added macroscopic background content in the introduction, including China's economic and social development during the study period. At the same time, we also readjusted the title of the paper according to the content, in order to better reflect the research content and main questions. The title is changed to “The allocation change of rural land consolidation structure under the influence factors of different geographical and economic development regional of China”

Point 3: Literature review section, lacks a critical Literature review should be concise rather than simply enumerating the literature.

Response 3: We have critical review of literature, summarized and refined the literature.

Point 4: 12 factors were selected from ecological, social and natural aspects, but didn't clarify why these factors are selected, are there any references to support this choice?

Response 4: We have noted the references, and analyzed the basis of factor selection.

Point 5: 3.1 data source section, the article lists nine land consolidation areas in China as the research area. A location map should be provided here with the location of the study area indicated.

Response 5: We have provided a location of the study area indicated.

  1. Regarding the methodology, the 3.3 transition index section states that the article uses the ratio of land development and land consolidation projects to measure the structure of land consolidation in China and reflect the function of land consolidation. Is there any literature to support this? If yes, please cite it.

Response 6: We have noted the references, and further explained the basis of measure the structure of land consolidation in China and reflect the function of land consolidation.

7.Figures: the figures resolution in part 4.1.2 is not enough resulting in blurriness and needs to be adjusted.

Response 7:We have replaced the relevant images.

Finally, special thanks to you for your constructive comments again!

Reviewer 3 Report

This paper has not met the basic line yet. The quality of the figures, the structure, and the contents are not standard. The English is very poor and difficult to read. I have to reject the paper and please prepare the manuscript carefully before submission. The comments are listed below to help prepare the manuscript:

1. It's already the year of 2023, why did the authors choose a study period of 2000 to 2014?

2. What are the TILC, QT, JY, FGH mean in the abstract?

3. Introduction: Many sentences lack of proper citations. No citation for the whole paragraph 2 and 3, it's unacceptable.

4. Line 43: "1929.18 *104 ha". Where does this number come from?

5. Literature Review: This part is confusing and not necessary. The authors should select the core contents and merge them into the Introduction.

6. Line 197: Which is the relevant research?

7. Again, no proper citation for the widely known analyzing method shown the Methodology chapter.

8. I only detected one citation in this whole chapter of "Results and Discussion". Without referring to the literature, how to make a trustful discussion?

9. Where did all the maps come from? No related contents in the method chapter.

10. The quality of all the figures is very low and hard to read. The authors should provide high dpi figures.

11. Many contents in chapter 4 and 5 are not from your own analysis. These contents are not related to your results shown in the first part of chapter 4. I have to suspect these contents because they don't have data support, and also, no citations.

Author Response

Manuscript ID: ijerph-2169335

Title: How does the land consolidation structure of China change under economic development in transformation period?

Journal: International Journal of Environmental Research and Public Health;

Section “Environmental Health”; 

Special Issue “Urban Land Use Planning and Evaluation of Environmental Carrying Capacity”.

List of Changes

First of all, we appreciate your careful review and positive comments! All of your comments are very important and really helpful to revise and improve our manuscript (MS). According to your comments, we have made relevant changes (words in red) to the whole MS.The detailed revisions are listed below, responding to your comments point by point.

  1. 3. Replies to reviewer #3

 Comments and Suggestions for Authors

This paper has not met the basic line yet. The quality of the figures, the structure, and the contents are not standard. The English is very poor and difficult to read. I have to reject the paper and please prepare the manuscript carefully before submission. The comments are listed below to help prepare the manuscript:

Point 1: It's already the year of 2023, why did the authors choose a study period of 2000 to 2014?

Response 1: Thank you very much for valuable comments. 

In May 2014, the Regulations on the Economical and Intensive Use of Land issued and implemented by the Ministry of Land and Resources defined the connotation and objectives of land consolidation. The scope of land consolidation in China is no longer limited to agricultural land or rural land, but has become the consolidation of the whole land.

Point 2ï¼› What are the TILC, QT, JY, FGH mean in the abstract?

Response 2:We have added the full name for the abbreviation that first appeared. TILC= is the index of land consolidation structure change; A_LA is the land arrangement area; A_LD is the land development area. The content of land consolidation should be carried out in space. The national land consolidation plan is a macroscopic land consolidation policy, which divides China into nine land consolidation areas. Land consolidation zoning is a comprehensive division of the region on the basis of considering the natural, social and economic, land use status, land use issues, land use direction, regional development strategy, and the integrity of provincial jurisdictional boundaries. Nine land consolidation zones in China are: Northeast China(NE), Jin-Yu(JY), Beijing-Tianjin-Hebei-Shandong(BTHL),Jiangsu-Zhejiang-Shanghai(JZS),Fujian-Guangdong-Hainan (FGH), Northwest China (NW), Southwest China (SW), Hunan-Hubei-Anhui-Jiangxi (HHAJ), Qinghai-Tibet (QT).

Point 3:  Introduction: Many sentences lack of proper citations. No citation for the whole paragraph 2 and 3, it's unacceptable.

Response 3: We have the appropriate references in the appropriate places.

Point 4: Line 43: "1929.18 *104 ha". Where does this number come from?

Response 4: The data comes from the Ministry of Natural Resources, People’s Republic of China (originally calledMinistry of Land and Resources of the People's Republic of China)

Point 5: Literature Review: This part is confusing and not necessary. The authors should select the core contents and merge them into the Introduction.

Response 5: We have rewritten the introduction, and re-selected and summarized the literature.

Point 6: Line 197: Which is the relevant research?

Response 6: We have marked the references

Point 7:  Again, no proper citation for the widely known analyzing method shown the Methodology chapter.

Response 7: We have marked the literature citations.

  1. I only detected one citation in this whole chapter of "Results and Discussion". Without referring to the literature, how to make a trustful discussion?

Response 8: We have marked the literature citations to make our discussion more comparative.

Point 9: Where did all the maps come from? No related contents in the method chapter.

Response 9: We have stated this in the data source, “China's vector map from the National Bureau of Surveying and Mapping Geographic Information”, ArcGIS 10.0 software platforms for map making, and land consolidation project data from the "China Land and Resources Statistics Yearbook".

Point 10: The quality of all the figures is very low and hard to read. The authors should provide high dpi figures.

Response 10: We've replaced it with a clearer picture(dpi 600)

Point 11: Many contents in chapter 4 and 5 are not from your own analysis. These contents are not related to your results shown in the first part of chapter 4. I have to suspect these contents because they don't have data support, and also, no citations.

Response 11: We reorganized and wrote the two parts, noted data sources and references. Land consolidation project data from the "China Land and Resources Statistics Yearbook".

Finally, special thanks to you for your constructive comments again!

Reviewer 4 Report

This article explores the spatio-temporal variation characteristics and driving factors of Land consolidation structure. It is rich in content, reasonable in method, and has good research significance. The main questions are:

(1) As the author said, the timeliness of the research is lacking, whether the selection of indicators is comprehensive and reasonable, the analysis effect at the provincial scale is poor, and the interpretation significance is limited. For these contents, the author can make some explanations in the data source and discussion. For example, why did you choose 2000-2014 as the research period? Differences across different research scales?

(2) The literature review is listed as a part of the content alone, and the introduction content can be shortened. It is recommended to condense the research objectives of this article in the introduction, and the literature review needs to be further improved. In addition, the reference format is unified, and some superscripts exist (line57).

(3) The basis for the selection of indicators, it is recommended to add relevant references.

(4) I am concerned that the English is not at the standard for publication. There are some mistakes throughout the manuscript (a lot of repeated phrase expressions and incoherent places). The manuscript should be checked by a native English speaker.

(5) The result analysis and discussion suggestions are separated, and the combination of the two makes the structure of the article more confusing.

(6) The clarity of the drawings needs to be improved and the rationality needs to be further considered. For example, the placement position of the legend (Figure 7, 8, 11, whether all the picture frames are connected together (Figure 12).

(7) To further streamline the conclusion, the last paragraph can be moved into the Discussion.

Author Response

Manuscript ID: ijerph-2169335

Title: How does the land consolidation structure of China change under economic development in transformation period?

Journal: International Journal of Environmental Research and Public Health;

Section “Environmental Health”; 

Special Issue “Urban Land Use Planning and Evaluation of Environmental Carrying Capacity”.

List of Changes

First of all, we appreciate your careful review and positive comments! All of your comments are very important and really helpful to revise and improve our manuscript (MS). According to your comments, we have made relevant changes (words in red) to the whole MS.The detailed revisions are listed below, responding to your comments point by point.

  1. 4. Replies to reviewer #4

 Comments and Suggestions for Authors

This article explores the spatio-temporal variation characteristics and driving factors of Land consolidation structure. It is rich in content, reasonable in method, and has good research significance. The main questions are:

Point 1: As the author said, the timeliness of the research is lacking, whether the selection of indicators is comprehensive and reasonable, the analysis effect at the provincial scale is poor, and the interpretation significance is limited. For these contents, the author can make some explanations in the data source and discussion. For example, why did you choose 2000-2014 as the research period? Differences across different research scales?

Response 1: Thank you very much for valuable comments. 

According to the land use zoning in the general land use planning outline, the existing land consolidation planning zones are divided into nine regions for project layout. The scale and structure of land consolidation in different regions should be arranged according to particular factors, which is an important issue concerning the effectiveness of land consolidation. It is of great significance to identify and analyze the influencing factors of each stage and region accurately, and then find out the driving mechanism by integrating multi-factors and multi-dimensions, and implement the guidance of differentiation in accordance with the scientific land consolidation management system.

In May 2014, the Regulations on the Economical and Intensive Use of Land issued and implemented by the Ministry of Land and Resources defined the connotation and objectives of land consolidation. The scope of land consolidation in China is no longer limited to agricultural land or rural land, but has become the consolidation of the whole land.

Point 2: The literature review is listed as a part of the content alone, and the introduction content can be shortened. It is recommended to condense the research objectives of this article in the introduction, and the literature review needs to be further improved. In addition, the reference format is unified, and some superscripts exist (line57).

Response 2: We have incorporated the literature review into the introduction, improved the literature review, condensed the research objectives, unified the reference format.

Point 3: The basis for the selection of indicators, it is recommended to add relevant references.

Response 3:  We have added the relevant references.

Point 4: I am concerned that the English is not at the standard for publication. There are some mistakes throughout the manuscript (a lot of repeated phrase expressions and incoherent places). The manuscript should be checked by a native English speaker.

Response 4: We apologize for the poor language of our manuscript. We worked on the manuscript for a long time and the repeated addition and removal of sentence and sections obviously led to poor readability.We have now worked on both language and readability and have also involved native English speakers for language corrections.We really hope that the flow and language level have been substantially improved.

Point 5: The result analysis and discussion suggestions are separated, and the combination of the two makes the structure of the article more confusing.

Response 5: We have separated the result analysis and discussion suggestions.

Point 6: The clarity of the drawings needs to be improved and the rationality needs to be further considered. For example, the placement position of the legend (Figure 7, 8, 11, whether all the picture frames are connected together (Figure 12).

Response 6:We've replaced it with a clear picture(dpi 600).  We redrew and adjusted the relevant images.

Point 7: To further streamline the conclusion, the last paragraph can be moved into the Discussion.

Response 7: We have moved the last paragraph into the discussion, streamline the conclusion.

Finally, special thanks to you for your constructive comments again!

Reviewer 5 Report

In my opinion, the part of the conclusions should be explained more.

Author Response

Manuscript ID: ijerph-2169335

Title: How does the land consolidation structure of China change under economic development in transformation period?

Journal: International Journal of Environmental Research and Public Health;

Section “Environmental Health”; 

Special Issue “Urban Land Use Planning and Evaluation of Environmental Carrying Capacity”.

List of Changes

First of all, we appreciate your careful review and positive comments! All of your comments are very important and really helpful to revise and improve our manuscript (MS). According to your comments, we have made relevant changes (words in red) to the whole MS.The detailed revisions are listed below, responding to your comments point by point.

  1. 5. Replies to reviewer #5

 Comments and Suggestions for Authors

In my opinion, the part of the conclusions should be explained more.

Response 1: Thank you very much for valuable comments. 

We have enriched the explanation of the part of the conclusion. The data, methods and main research content of this paper are introduced.

Finally, special thanks to you for your constructive comments again!

Round 2

Reviewer 2 Report

The revised version has addressed what my concerns, I suggest accepting in current form.

Author Response

Manuscript ID: ijerph-2169335

Title: How does the land consolidation structure of China change under economic development in transformation period?

Journal: International Journal of Environmental Research and Public Health;

Section “Environmental Health”; 

Special Issue “Urban Land Use Planning and Evaluation of Environmental Carrying Capacity”.

List of Changes

First of all, we appreciate your careful review and positive comments! All of your comments are very important and really helpful to revise and improve our manuscript (MS). According to your comments, we have made relevant changes (words in red) to the whole MS.The detailed revisions are listed below, responding to your comments point by point.

  1. 2. Replies to reviewer #2

Comments and Suggestions for Authors

The revised version has addressed what my concerns, I suggest accepting in current form.

Response 1: Thank you very much for valuable comments. 

Finally, special thanks to you for your constructive comments again!

Reviewer 3 Report

The manuscript has been significantly improved, thank you for your effort.

Author Response

Manuscript ID: ijerph-2169335

Title: How does the land consolidation structure of China change under economic development in transformation period?

Journal: International Journal of Environmental Research and Public Health;

Section “Environmental Health”; 

Special Issue “Urban Land Use Planning and Evaluation of Environmental Carrying Capacity”.

List of Changes

First of all, we appreciate your careful review and positive comments! All of your comments are very important and really helpful to revise and improve our manuscript (MS). According to your comments, we have made relevant changes (words in red) to the whole MS.The detailed revisions are listed below, responding to your comments point by point.

  1. 2. Replies to reviewer #3

Comments and Suggestions for Authors

The manuscript has been significantly improved, thank you for your effort.

Response 1: Thank you very much for valuable comments. 

Finally, special thanks to you for your constructive comments again!

Reviewer 4 Report

The authors have carefully revised the manuscript according to my suggestions. However, some minor flaws make the article needed to be revised before it can be considered for publication.

(1) There are still many repetitive expressions in the article, such as "In this paper" appearing many times, which needs further polishing.

(2) There are too many paragraphs in the introduction, line 123 is already summarizing, and then there are three paragraphs later, it is recommended that the author further cut and condense the content of this part.

(3) All the  figures are not very clear, and some figures are merged, but the sub- figures are not aligned, as shown in Figure 1, Figure 11 and Figure 12.

(4) Further reduce and enhance the discussion content.

Author Response

Manuscript ID: ijerph-2169335

Title: How does the land consolidation structure of China change under economic development in transformation period?

Journal: International Journal of Environmental Research and Public Health;

Section “Environmental Health”; 

Special Issue “Urban Land Use Planning and Evaluation of Environmental Carrying Capacity”.

List of Changes

First of all, we appreciate your careful review and positive comments! All of your comments are very important and really helpful to revise and improve our manuscript (MS). According to your comments, we have made relevant changes (words in red) to the whole MS.The detailed revisions are listed below, responding to your comments point by point.

  1. 2. Replies to reviewer #4

Comments and Suggestions for Authors

The authors have carefully revised the manuscript according to my suggestions. However, some minor flaws make the article needed to be revised before it can be considered for publication.

Point 1: There are still many repetitive expressions in the article, such as "In this paper" appearing many times, which needs further polishing.

Response 1: Thank you very much for valuable comments. 

We are very sorry for that. We have re-checked and modified the repetitive expressions and adjusted the similar parts of the full text.

Point 2: There are too many paragraphs in the introduction, line 123 is already summarizing, and then there are three paragraphs later, it is recommended that the author further cut and condense the content of this part.

Response 2: You're right. We've streamlined and reorganized the introduction.

Point 3: All the  figures are not very clear, and some figures are merged, but the sub- figures are not aligned, as shown in Figure 1, Figure 11 and Figure 12.

Response 3: We redrew and replaced some of the images.

Point 4: Further reduce and enhance the discussion content.

Response 4: We have made targeted simplification and adjustments to the discussion section, making it more correspond to the previous part of the article and comparable to the current research.

Finally, special thanks to you for your constructive comments again!